

# Fluid resuscitation-related coagulation impairment in a porcine hemorrhagic shock model

Alexander Ziebart, Robert Ruemmler, Christian Möllmann, Jens Kamuf, Andreas Garcia-Bardon, Serge C. Thal and Erik K. Hartmann

Department of Anesthesiology, Medical Centre of the Johannes Gutenberg-University, Mainz, Germany

## ABSTRACT

**Background**. Fast and effective treatment of hemorrhagic shock is one of the most important preclinical trauma care tasks e.g., in combat casualties in avoiding severe end-organ damage or death. In scenarios without immediate availability of blood products, alternate regimens of fluid resuscitation represent the only possibility of maintaining sufficient circulation and regaining adequate end-organ oxygen supply. However, the fluid choice alone may affect the extent of the bleeding by interfering with coagulation pathways. This study investigates the impact of hydroxyethyl starch (HES), gelatine-polysuccinate (GP) and balanced electrolyte solution (BES) as commonly used agents for fluid resuscitation on coagulation using a porcine hemorrhagic shock model.

**Methods**. Following approval by the State and Institutional Animal Care Committee, life-threatening hemorrhagic shock was induced via arterial blood withdrawal in 24 anesthetized pigs. Isovolumetric fluid resuscitation with either HES, GP or BES ($n = 3 \times 8$) was performed to compensate for the blood loss. Over four hours, hemodynamics, laboratory parameters and rotational thromboelastometry-derived coagulation were analyzed. As secondary endpoint the porcine values were compared to human blood.

**Results**. All the agents used for fluid resuscitation significantly affected coagulation. We measured a restriction of laboratory parameters, clot development and clot firmness, particularly in HES- and GP-treated animals. Hemoglobin content dropped in all groups but showed a more pronounced decline in colloid-treated pigs. This effect was not maintained over the four-hour monitoring period.

**Conclusion**. HES, GP, and BEL sufficiently stabilized the macrocirculation, but significantly affected coagulation. These effects were most pronounced after colloid and particularly HES administration. Despite suitability for rapid hemodynamic stabilization, colloids have to be chosen with caution, because their molecular properties may affect coagulation directly and as a consequence of pronounced hemodilution. Our comparison of porcine and human coagulation showed increased coagulation activity in pig blood.

Corresponding author
Alexander Ziebart,
alexander.ziebart@unimedizin-mainz.de

## BACKGROUND

Massive blood loss as a hemorrhagic shock causes requires resolute and efficient action to prevent severe injury or death. As a factor in 15% of all fatal injuries, it is one of the most challenging issues in prehospital and early clinical scenarios (*Kutcher et al., 2013*; *Noll et al., 2017*). The instability that results from massive blood loss induces an undersupply of oxygen and can cause irreversible organ failure (*Nielsen et al., 2014*; *Dubniks, Persson & Grande, 2009*). To combat this problem, different regimens and guidelines focusing on fluid resuscitation have been established (*Kozek-Langenecker et al., 2013*; *Spahn et al., 2019*). For years, colloids and especially hydroxyethyl starch (HES) have been indispensable for nearly all forms of shock resulting from relative or absolute hypovolemia. Their efficacy relies on macromolecule-related increases in oncotic pressure and decreased extravasation. Crystalloids, however, act only temporarily due to a free volume-specific shift into the extravascular space (*Orbegozo Cortes et al., 2015*). Several clinical studies have questioned the safety of HES in the treatment of critical ill patients (*Guidet et al., 2012*; *Brunkhorst et al., 2008*; *Perner et al., 2012*). Side effects such as increased risk of acute kidney injury, which requires renal replacement therapy and can even lead to mortality, have challenged the role of HES in the intensive care unit (*Annane et al., 2013*; *Dellinger et al., 2013*; *Myburgh et al., 2012*; *Ferreira et al., 2018*). Other detrimental effects of HES include increased perioperative blood loss and coagulopathy. Some studies, however still regard HES administration as reliable option, particularly in isolated, manageable bleeding scenarios and patient without risk constellation for acute kidney injury, which has led to controversial discussions (*Rehm et al., 2019*; *Gillies et al., 2014*). In the past, the European Medicines Agency (EMA) recommended avoiding HES and prohibited its use in patients with severe sepsis or burn injuries (*Spahn et al., 2019*). In 2018 and 2019, however, the EMA re-permitted administration of HES for the management of acute blood loss-induced hypovolemia while pointing out several contraindications, including sepsis, severe coagulopathy and renal impairment or replacement therapy (*Spahn et al., 2019*; *EMA, 2018*). Nevertheless, these recommendations are undermined by lack of alternative solutions and clinical scenarios that represent a grey zone between defined indications and prohibitions. Gelatine-polysuccinate (GP) represents an alternative but less frequently applied colloid based on bovine collagen. Apart from its shorter intravascular persistence, GP appears to be comparable to modern HES 130/0.4 (*Beyer et al., 1997*; *Awad et al., 2012*). Interestingly, knowledge of the disadvantages and advantages of GP is limited (*Qureshi et al., 2016*; *National Clinical Guideline C, 2013*; *Wu et al., 2015*; *Witt et al., 2016*), with the exception of a higher risk of anaphylaxis compared to other colloids (*Moeller et al., 2016*; *Ertmer et al., 2009*). Bleeding control is a crucial task during hemorrhagic shock and requires sufficient coagulation maintenance. Though indicated in blood loss-related hypervolemia, HES bears the risk of coagulation impairment. (*Rasmussen, Secher & Pedersen, 2016*; *Entholzner et al., 2000*; *Stump et al., 1985*; *Reuteler et al., 2017*; *Sanfelippo, Suberviola & Geimer, 1987*; *Treib et al., 1996*). Comparisons between HES and GP have shown similar detrimental effects, but only few studies have investigated these effects during hemorrhagic shock (*Witt et al., 2012*; *Haas et al., 2008*; *Mittermayr et al., 2007*).

Our aim was to explore the influence of HES and GP on coagulation with a focus on point-of-care rotational thromboelastometry (ROTEM), which enables to fibrin polymerization analysis, clot firmness and formation, platelet function and fibrinolysis through different assays (*Entholzner et al., 2000*). Viscoelastic point-of-care methods are explicitly discussed in the European guidelines for the management of traumatic and perioperative bleeding (*Spahn et al., 2019*; *Durila et al., 2018*) and are an important instrument in early goal-directed coagulation therapy (*Tanczos, Nemeth & Molnar, 2015*). We hypothesized that comparable amounts of HES and GP would impair the coagulation cascade more than balanced electrolyte solution (BES) in a pig model simulating a preclinical hemorrhagic shock scenario.

## METHODS

This study was approved by the State and Institutional Animal Care Committee (Landesuntersuchungsamt Rheinland-Pfalz, Koblenz, Germany; Chairperson: Dr Silvia Eisch-Wolf; reference number: 23 177-07/G 15-1-092; 01/2016) in accordance with the ARRIVE guidelines. This study represents an independent sub-project and complementary hypothesis of a research project that investigates the cerebral effects of solutions for fluid resuscitation and did not increase the overall number of animal experiments within this project. We included 24 juvenile male pigs (*sus scrofa domestica*; mean weight $28 \pm 2$ kg; age: 8–12 weeks) in a prospective randomized animal experiment.

### Anesthesia and instrumentation

The anesthesia, instrumentation and shock induction were performed as previously described (*Ziebart et al., 2018*; *Ziebart et al., 2019*). The animals stayed in their known environment for as long as possible to minimize stress, and breeder controlled environmental conditions. The animals were sedated via intramuscular injections of ketamine (8 mg/kg) and midazolam (0.2 mg/kg) to minimize their stress during transport. Fentanyl (4 µg/kg), propofol (4 mg/kg) and atracurium (0.5 mg/kg) were injected intravenously via an ear-vein for general anesthesia and endotracheal intubation. Continuous infusion of fentanyl (0.1–0.2 mg/kg/h) and propofol (8–12 mg/kg/h) maintained the anesthesia. During the experiment, the animals were held under general anesthesia by means of continuous propofol (8–12 mg/kg/h) and fentanyl (0.1–0.2 mg/kg/h) and ventilated in the following volume-controlled mode (Evita 2, Draeger, Lübeck, Germany): positive end-expiratory pressure of 5 cmH$_2$O, a tidal volume of 8 ml/kg, fraction of inspired oxygen of 0.4, an inspiration to expiration ratio of 1:2 and a variable respiration rate to achieve end-tidal CO$_2$ <6 kPa.

Four femoral vascular catheters were placed using ultrasound guidance as follows: central venous line (drug administration and thermodilution-related cold saline application, left femoral vein), pulse contour cardiac output system (PiCCO, Pulsion Medical Systems, Germany, right femoral artery), arterial line (blood withdrawal for shock model, left femoral artery) and large-bore venous introducer (fluid resuscitation, right femoral vein). Hemodynamic and spirometric parameters were constantly measured and recorded (LabVIEW, AD-Instruments, Sydney, Australia).
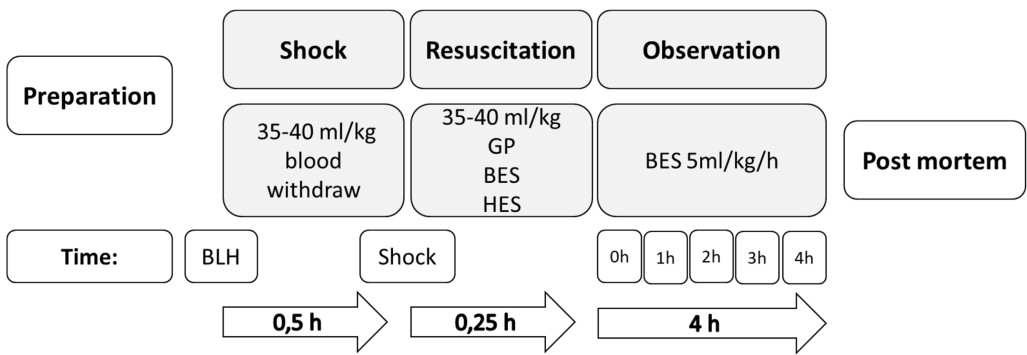

**Figure 1** Experimental flow chart. GP, gelatine-polysuccinate; HES, hydroxyethyl starch; BES, balanced electrolyte solution; BLH, baseline healthy.

## Experimental design

Instrumentation was performed, followed by 30 min of consolidation and baseline documentation. Hemorrhagic shock was induced via arterial blood withdrawal (35–40 ml/kg) over 15 min until mean arterial pressure (<45 mmHg) and cardiac index (<40% of the baseline value) decreased (*Ziebart et al., 2019*). Figure 1 displays a short summary of this protocol.

Thirty minutes after the shock induction, the animals were randomized and treated according to a group-specific fluid resuscitation regime and adjusted, according to the volume of removed blood, as follows:

GP (Gelafundin iso 4%, B. Braun, Germany; $n = 8$)
HES 130/0.4 (Volulyte 6%, Fresenius Kabi AG, Germany; $n = 8$)
BES (Sterofundin iso, B. Braun, Germany; $n = 8$)

No specific cardio-circulatory parameters influenced this approach. No further fluids or catecholamine were administered other than a permanent BES infusion (5 ml/kg/h). At the baseline, following fluid resuscitation and at the end of the experiment (baseline, 0 h and 4 h), blood was withdrawn for coagulation and laboratory analysis into standard test tubes. After four hours of monitoring, the animals were killed during deep general anesthesia via central venous administration of propofol and potassium chloride.

## Laboratory assessment

Blood samples were collected at the baseline, following fluid resuscitation (0 h) and after four hours of observation (4 h). The blood was stored in tubes with 0.14 ml citrate solution (S-Monovette®, Sarstedt, Nuembrecht, Germany). Laboratory blood tests, including partial thromboplastin time (PTT), prothrombin time (PT), fibrinogen, platelet count, hemoglobin content and hematocrit, were used to analyze standard coagulation and blood parameters.

## Rotational thromboelastometry

Rotational thromboelastometry (ROTEM® delta; TEM International GmbH, Munich, Germany) analyses were performed according to operation instructions. Thromboelastometry was able to detect changes in the viscoelastic characteristics of a citrate blood clot. The blood samples were mixed with specific reagents to allow for analysis of coagulation properties through the following five tests: EXTEM, which investigates the tissue factor pathway (extrinsic); INTEM, which determines the contact activation pathway (intrinsic); HEPTEM, which analyses heparin-induced effects in comparison to INTEM; FIBTEM, which examines fibrinogen deficiency or restricted fibrin polymerization in clots through the addition of cytochalasin D to inhibit platelet contribution; and APTEM, which uses aprotinin to analyze fibrinolysis. We performed all tests excluding HEPTEM, as no heparin was provided in the study protocol. To validate the measured values, we compared the baseline parameters with human references (*Lang et al., 2005*). Thromboelastometry analyses depict three phases of coagulation: clot formation (clotting time = CT, time until clot starts to form; clot formation time = CFT, time between CT and an amplitude of 20 mm clot firmness; alpha-angle = alpha, angle between 0 and 2 mm of amplitude), clot firmness (A10 and A20, amplitude of clot firmness at 10 and 20 min; maximum clot firmness = MCF, the absolute highest amplitude of the curve) and lysis (maximum lysis = ML, maximum percentage loss of clot strength compare to MCF) (*Mauch et al., 2013*; *Prat et al., 2017*).

## Statistics

We displayed the data in terms of mean and standard deviation (SD). The intergroup effects were analyzed using two-way analysis of variance (ANOVA). The Student-Newman-Keuls method was used for pairwise multiple comparison corrections. $P$ values lower than 5% were considered significant. The statistical analysis was performed utilizing software package SigmaPlot 12.5 (Systat Software). The porcine reference values were displayed as confidence intervals of a confidence level of 95%.

## RESULTS

Comparable amounts of blood were withdrawn ($33 \pm 5$ ml/kg) for shock induction from all three groups. Arterial blood pressure ($58 \pm 11\%$) and cardiac index ($50 \pm 12\%$) significantly decreased in a shock-like fashion versus the baseline (each $p < 0.01$) (Table 1). The amounts of fluid administered were also comparable.

In the HES and BES groups, a survival rate of 100% was achieved. In the GP group, one animal died during a tachyarrhythmia episode one hour after the fluid resuscitation. The different infusion regimens led to adequate hemodynamic stabilization in all groups. The cardiac index increase was significantly greater following colloid infusion. After four hours, all values returned to the baseline level, with the exception of heart rate in the BES group (Table 1). Immediately following shock induction, the hemoglobin content was relatively unaltered; however, it significantly decreased throughout fluid resuscitation, with a pronounced decrease in the colloid groups (each $p < 0.01$). Fibrinogen, PT and

**Table 1** Extended hemodynamic and blood parameters.

| Parameter | | BLH | Shock | 0 h | 2 h | 4 h |
|---|---|---|---|---|---|---|
| **MAP** | HES | 66 (8) | 31 (10)[*] | 80 (11)[*] | 67 (8) | 65 (7) |
| | GP | 70 (9) | 25 (5)[*] | 76 (16) | 59 (10)[*] | 61 (6)[*] |
| | BES | 61 (11) | 29 (4)[*] | 73 (10) | 58 (8) | 58 (9) |
| **HR** | HES | 75 (17) | 116 (38)[*] | 123 (42)[*] | 99 (33) | 99 (33) |
| | GP | 84 (10) | 145 (53)[*] | 137 (26)[*] | 106 (27) | 104 (21) |
| | BES | 78 (22) | 144 (46)[*] | 118 (31)[*] | 127 (48)[*] | 134 (40)[*] |
| **CI** | HES | 4.2 (0.7) | 2.1 (0.4)[*] | 7.5 (2.6)[*#3] | 5.7 (1.4)[*] | 4.7 (0.8) |
| | GP | 4.8 (1) | 2 (0.3)[*] | 7.6 (2)[*#2] | 5.4 (1.3) | 5.6 (1.6) |
| | BES | 4.1 (1.2) | 2.2 (0.4)[*] | 5.4 (0.9)[*#2,3] | 4.4 (0.6) | 4.7 (0.6) |
| **HCT** | HES | 26.6 (2.5) | 23.2 (1.7) | 13.4 (2)[*#3] | 16.1 (3.2)[*#3] | 17 (2.1)[*] |
| | GP | 28.5 (1.4) | 23 (2.1) | 11.9 (1.7)[*#2] | 18.2 (0.5)[*#2] | 14.2 (1.6)[*] |
| | BES | 26.1 (2.3) | 23.6 (2.5) | 17 (1.8)[*#2,3] | 19.6 (1.4)[*#2,3] | 16.8 (1)[*] |
| **HB** | HES | 8.6 (0.8) | 7.5 (0.5) | 4.3 (0.6)[*#3] | 5 (0.7)[*#3] | 5.1 (0.7)[*] |
| | GP | 9.2 (0.5) | 7.5 (0.7) | 3.8 (0.6)[*#2] | 4.7 (0.9)[*#2] | 4.7 (0.5)[*] |
| | BES | 8.5 (0.7) | 7.7 (0.8) | 5.5 (0.5)[*#2,3] | 5.5 (0.3)[*#2,3] | 5.4 (0.3)[*] |

**Notes.**
[*]indicates $p < 0.05$ timepoint comparison to baseline.
[#]indicates $p < 0.05$ in intergroup comparison (1 = HES vs. GP; 2 = GP vs. BES; 3 = HES vs. BES).
SD, Standard deviation; MAP, mean arterial pressure (mmHg); HR, heart rate ($min^{-1}$); CI, cardiac index [$l/min/m^2$]; Hct, Hematocrit (%); Hb, Hemoglobin (g/dl).

PTT became considerably impaired in the colloid groups, whereas the BES group was only affected to a minor extent (Table 1 and Fig. 2).

The rotational thromboelastometry data showed no intergroup differences at the baseline. Parameters concerning the temporal sequence (CT, CFT, alpha) as well as clot firmness (A10, A20, MCF) were significantly impaired following administration of GP and HES, particularly in EXTEM but also in INTEM. HES exerted the most detrimental effects, followed by GP. The BES group varied significantly from the baseline, as-well, but were still higher compared to the colloid groups (Figs. 3 and 4). Similar to EXTEM significant impairment of FIBTEM and APTEM in the colloid groups was observed (Figs. 5 and 6). Regrettably, the FIBTEM clotting time in the GP and HES groups were indeterminable due to technical errors.

All rotational thromboelastometry-derived parameters recovered over the four hours in all groups, but the extent of recovery differed between the different regimes. In both EXTEM and APTEM, CT and CFT were most prolonged through HES administration compared to the baseline. and to the other groups. CFT returned to the baseline level only in the BES group (Fig. 3 & Fig. 6). Amplitude and MCF were restricted in all groups but nearly reached the baseline level in the BES-treated animals (Figs. 3 and 4). INTEM results were not as pronounced as the EXTEM, FIBTEM and APTEM results. We observed no

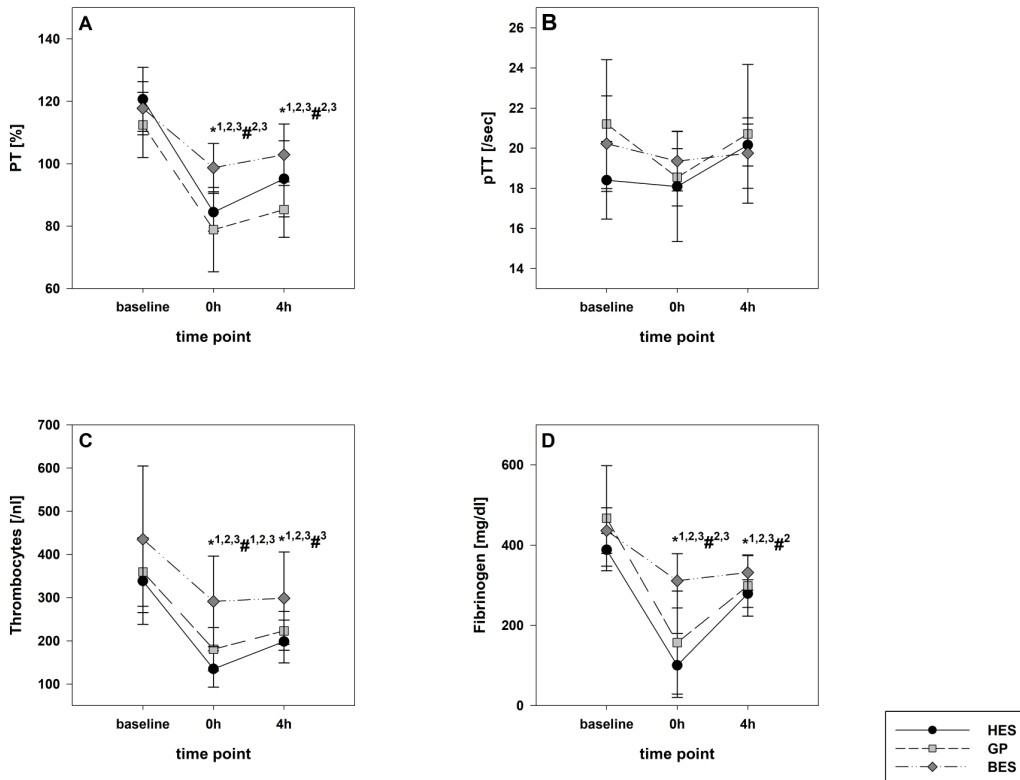

**Figure 2 Laboratory coagulation parameters.** (A) Prothrombin time; (B) activated partial thromboplastin time; (C) platelet count; (D) fibrinogen Group effects over time were analyzed using two-way ANOVA and the post-hoc Student-Newman-Keuls method. $*$, $p < 0.05$ timepoint comparison to the baseline (1, HES; 2, GP; 3, BES); #, $p < 0.05$ in the intergroup comparison (1, HES vs. GP; 2, GP vs. BES; 3, HES vs. BES); GP, gelatine-polysuccinate; HES, hydroxyethyl starch; BES, balanced electrolyte solution.

restrictions in ML and did not detect any signs of hyperfibrinolysis in any of the groups (Figs. 3 and 6).

Our comparison of porcine and human coagulation showed increased coagulation activity in pig blood. This effect could be observed in all parameters and involved decreased clotting times and increased amplitudes (Table 2).

# DISCUSSION

This study investigated the influence of HES, GP and BES on coagulation during early hemorrhagic shock in a pig model, with a focus on rotational thromboelastometry. Accordingly, we found an impaired coagulatory activity that concerns temporal kinetics as well as clot firmness as a consequence of fluid resuscitation in all groups, but with the most deleterious effects through HES administration followed by GP. We simulated a frequent preclinical or clinical blood loss scenario with subsequent bleeding control (i.e., during surgery). Shock was induced by defined blood loss in combination with hemodynamic deterioration. If untreated, this model leads to cardio-circulatory failure and death in 66% of the animals within two hours (*Ziebart et al., 2019*; *Ziebart et al., 2018*).

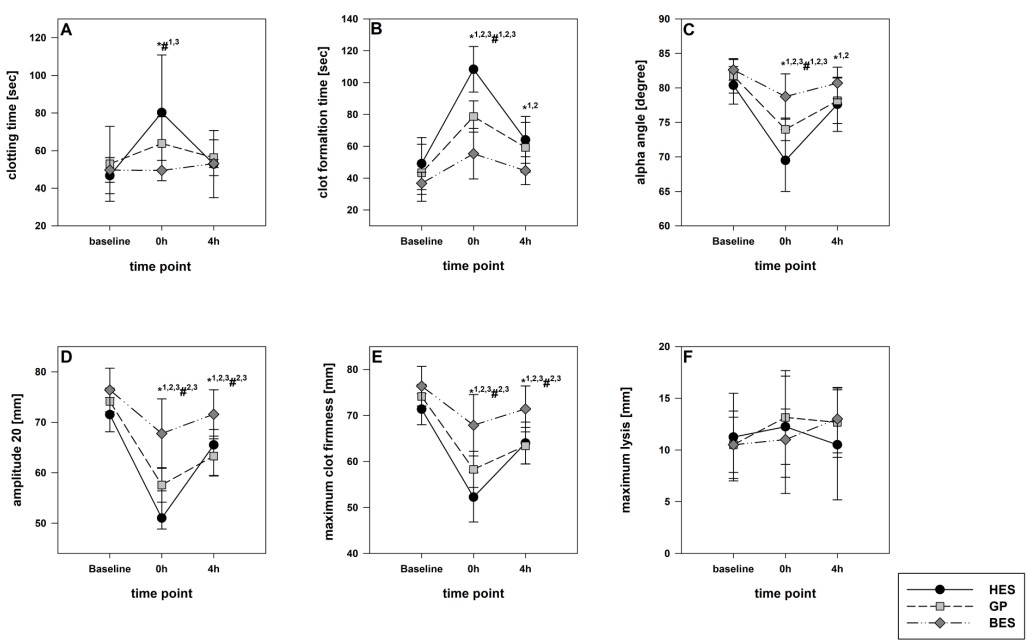

**EXTEM**

**Figure 3 EXTEM (tissue factor pathway).** (A) Clotting time; (B) clot formatting time; (C) amplitude 20; (D) maximum clot firmness Group effects over time were analyzed using two-way ANOVA and the post-hoc Student-Newman-Keuls method. *, $p < 0.05$ timepoint comparison to the baseline (1, HES; 2, GP; 3, BES); #, $p < 0.05$ in the intergroup comparison (1, HES vs. GP; 2, GP vs. BES; 3, HES vs. BES); GP, gelatine-polysuccinate; HES, hydroxyethyl starch; BES, balanced electrolyte solution.

In the scenario sufficient and quick action is necessary, and ongoing fluid therapy will not stop immediately after bleeding control. Hemodynamic values were allowed to fall below the acceptable thresholds of permissive hypotension (*Spahn et al., 2019*) Isovolemic amounts of fluids were applied using a 1:1 replacement regime. The chosen group size was adjusted on account of comparable large animal studies, that also focused on hemorrhagic shock or cardiorespiratory effects (*Ziebart et al., 2018*; *Ziebart et al., 2014*).The scenario concentrated on rapid stabilization to avoid the undersupply of oxygen to the vital organs and did not stop immediately after the bleeding was controlled.

Sufficient hemodynamic recovery through fluid resuscitation was registered for all three regimens. In the colloid groups particularly, the baseline level was surpassed. After four hours, we did not detect any significant intergroup differences, with the exception of heart rate, which remained increased in the GP and BES groups. These results were expected due to the high effectivity of colloids in increasing osmotic pressure leading to increased intravascular volume. A ratio of 1:1.5 between colloids and crystalloids would theoretically achieve similar hemodynamics but would not have simulated a realistic clinical scenario (*Annane et al., 2013*). As expected, we observed significantly lower hemoglobin content following the HES and GP administration, a sign of hemodilution.

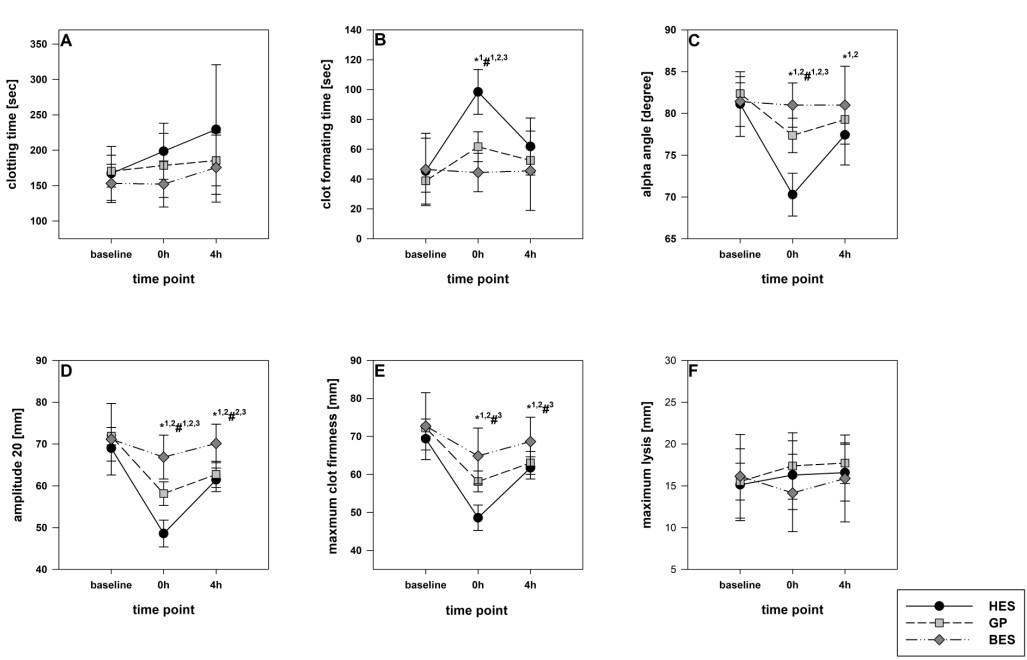

**INTEM**

**Figure 4   INTEM (contact activation pathway).** (A) Clotting time; (B) clot formatting time; (C) amplitude 20; (D) maximum clot firmness Group effects over time were analyzed using two-way ANOVA and the post-hoc Student-Newman-Keuls method. *, $p < 0.05$ timepoint comparison to the baseline (1, HES; 2, GP; 3, BES); #, $p < 0.05$ in the intergroup comparison (1, HES vs. GP; 2, GP vs. BES; 3, HES vs. BES); GP, gelatine-polysuccinate; HES, hydroxyethyl starch; BES, balanced electrolyte solution.

Hemodilution partially explains the impaired coagulation in the laboratory parameters and thromboelastometry results. Conventional coagulation parameters, clot formatting and firmness decreased in all groups, with the highest restriction following HES administration, then GP. These effects were more expressed in the extrinsic pathway, fibrinogen content and platelet count. Additionally, we compared our porcine baseline values with published human reference values; the porcine coagulation activity seemed to be higher than that of humans. This can especially be seen in the decreased clotting and clot formatting time and the pronounced clot firmness among the porcine specimens. The related results in FIBTEM still after four hours can be explain by a lack of fibrinogen that is not displayed into the conventional coagulation parameters. These findings correspond with results from other studies in which porcine hypercoagulation was described (*Lechner et al., 2018*; *Velik-Salchner et al., 2006*). These findings must be considered when interpreting the results in order for the right conclusions from this model to be drawn.

As described above, crystalloids are less effective (25%) due to fluid dispersion into the interstitium (*Orbegozo Cortes et al., 2015*; *Zaar et al., 2009*). Zaar et al. presented the consequences of hemodilution and demonstrated exacerbated liver bleeding following HES administration, whereas crystalloids mitigated the effect (*Zaar et al., 2009*). Other studies

**FIBTEM**

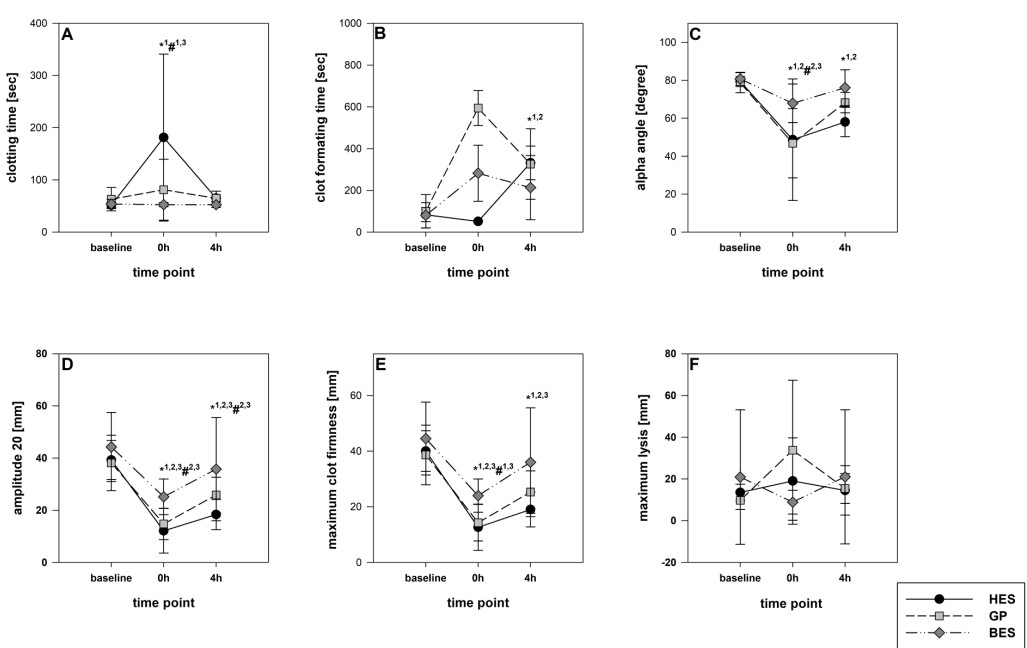

**Figure 5**  **FIBTEM-Test (fibrinogen deficiency detection).** (A) Clotting time; (B) clot formatting time; (C) amplitude 20; (D) maximum clot firmness Group effects over time were analyzed using two-way ANOVA and the post-hoc Student-Newman-Keuls method. *, $p < 0.05$ timepoint comparison to the baseline (1, HES; 2, GP; 3, BES); #, $p < 0.05$ in the intergroup comparison (1, HES vs. GP; 2, GP vs. BES; 3, HES vs. BES); GP, gelatine-polysuccinate; HES, hydroxyethyl starch; BES, balanced electrolyte solution.

have claimed that colloids affect coagulation by themselves (*Mauch et al., 2013*; *Dirkmann et al., 2008*). We observed similar results in our study directly after fluid resuscitation. However, intergroup differences were temporally limited, as hemoglobin content and hematocrit remained unchanged after four hours. Hemodilution reasonably explains coagulation deterioration directly following fluid resuscitation, but not its long-term effects. Several mechanisms have been described to explain this coagulopathy through colloids, including the negative influences of factor VIII (*Stump et al., 1985*), factor XIII, fibrinolysis (*Reuteler et al., 2017*), the von Willebrand factor (*Sanfelippo, Suberviola & Geimer, 1987*) and surface receptor GPIIb/IIIa on platelets (*Treib et al., 1996*). Administration of HES can particularly trigger these effects, which are pronounced by different types of HES. In this context, negative side effects of the new HES generation (130/0.4) are less relevant than those of older ones (e.g., HES 200/0.5), in which coagulation is impaired to a higher degree (*Entholzner et al., 2000*). Mauch et al. reported on reduced fibrin polymerization, which depends on decreased factor XIII activity for restricted clot firmness (*Mauch et al., 2013*; *Dirkmann et al., 2008*). This corresponds with our thromboelastometry results, including the decreases in MCF, which can be interpreted as a sign of impaired fibrin cross-linking in the colloid groups (e.g., Fig. 3) (*Reuteler et al., 2017*). We also detected

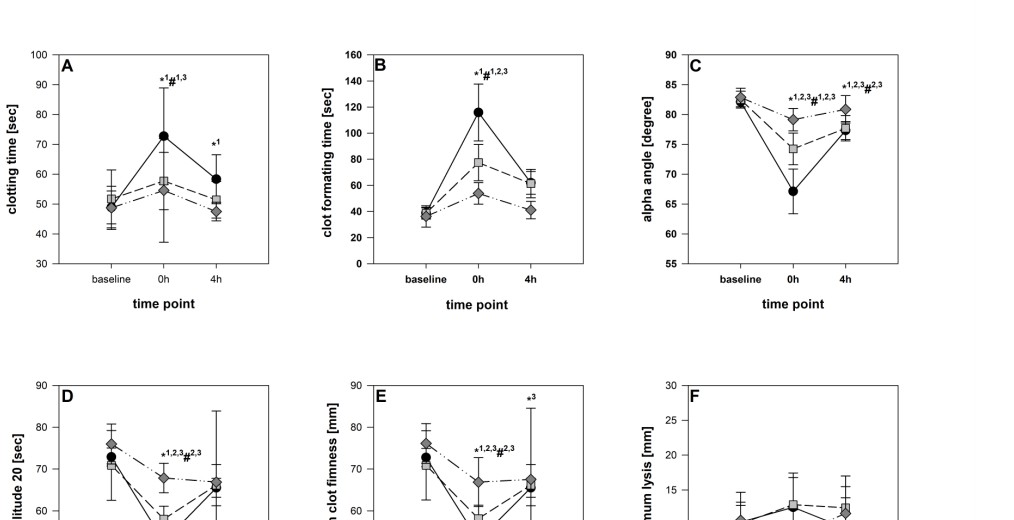

**APTEM**

**Figure 6  APTEM-Test (hyper-fibrinolysis-detection).** (A) Clotting time; (B) clot formatting time; (C) amplitude 20; (D) maximum clot firmness. Group effects over time were analyzed using two-way ANOVA and the post-hoc Student-Newman-Keuls method. *, $p < 0.05$ timepoint comparison to the baseline (1, HES; 2, GP; 3, BES); #, $p < 0.05$ in the intergroup comparison (1, HES vs. GP; 2, GP vs. BES; 3, HES vs. BES); GP, gelatine-polysuccinate; HES, hydroxyethyl starch; BES, balanced electrolyte solution.

further decreases in CT and alpha-angle as an indication of impaired clot development related to factor VIII and the von Willebrand factor. These results are in line with Haas et al., who reported compromised clot firmness following HES administration compared to GP in pigs (*Haas et al., 2008*). This effect appears to be dose-dependent (*Witt et al., 2012*). Furthermore, the direct influence of HES on thrombocyte function, which impairs rolling and adhesion capabilities, restricts GPIIb/IIIa receptor binding and direct degeneration through increased osmotic pressure (*Reuteler et al., 2017*; *Treib et al., 1996*). In theory, an effect of colloids on fibrinolysis is possible, but we did not detect any signs of hyperfibrinolysis. Lower degrees of hyperfibrinolysis may exceed the detection ability of a rotational thromboelastometry assay. The minor sensitivity to detect beginning hyperfibrinolysis and the time consuming analysis is a relevant limitation in emergency situations (*Larsen et al., 2012*). As another item of our study, we compared the measured porcine baseline values with standard parameters for human assays. We demonstrated that human assessments are suitable for this kind of research and generate useful values with general increased porcine coagulation (Table 2) (*Lang et al., 2005*). This might also help to established regimes for fluid resuscitation that can used likewise in veterinary medicine, where the number of relevant studies is limited (*Haas et al., 2008*; *Wong & Koenig, 2017*).

Ziebart et al. (2020), PeerJ, DOI 10.7717/peerj.8399

Peerj

**Table 2  Comparison of human and porcine thromboelastometric references ($n = 24$).** Data are displayed in the 95% confidence interval of absolute values (AV) and in relation (R) (displayed in percentage) to human values.

| TEST | EXTEM | | | | INTEM | | | | FIBTEM | | | | APTEM | | | |
|---|---|---|---|---|---|---|---|---|---|---|---|---|---|---|---|---|
| SPECIES | HUMAN | | PIG | | HUMAN | | PIG | | HUMAN | | PIG | | HUMAN | | PIG | |
| UNIT | AV | R | AV | R | AV | R | AV | R | AV | R | AV | R | AV | R | AV | R |
| CT | 42-74 | 100 | 45-55 | 86 | 137-246 | 100 | 149-207 | 93 | 43-69 | 100 | 50-62 | 100 | 42-74 | 100 | 47-53 | 86 |
| CFT | 46-148 | 100 | 36-46 | 42 | 42-100 | 100 | 36-51 | 61 | 46-148 | 100 | 60-107 | 86 | 46-148 | 100 | 36-40 | 39 |
| ALPHA | 63-81 | 100 | 81-83 | 114 | 71-82 | 100 | 81-83 | 106 | 63-81 | 100 | 78-81 | 110 | 63-81 | 100 | 82-83 | 115 |
| A10 | 43-65 | 100 | 70-74 | 133 | 44-68 | 100 | 68-72 | 125 | 9-24 | 100 | 50-62 | 339 | 43-65 | 100 | 69-74 | 132 |
| A20 | 50-69 | 100 | 72-76 | 124 | 50-71 | 100 | 69-73 | 117 | 8-21 | 100 | 36-45 | 279 | 50-69 | 100 | 71-76 | 124 |
| MCF | 49-71 | 100 | 72-76 | 123 | 52-72 | 100 | 69-73 | 115 | 9-25 | 100 | 37-45 | 241 | 49-71 | 100 | 71-76 | 123 |
| ML | 0-18 | 100 | 9-12 | 117 | 0-12 | 100 | 14-17 | 258 | 13-27 | 100 | 9-13 | 55 | 0-18 | 100 | 9-11 | 111 |

**Notes.**

CT, clotting time (sec); CFT, clot formation time (sec); APLHA, alpha-angle (degree); A10, amplitude 10% (mm); A20, amplitude 20% (mm); MCF, Maximum clot firmness (mm); ML, maximum lysis (mm).

In recent years, use of gelatine-based colloids has increased due to the unclear status of HES. However, beneath the risk of anaphylaxis and kidney injury, evidence concerning the risk profile of GP remains scarce (*Moeller et al., 2016*; *Demir et al., 2015*). Fast and effective therapy for hemorrhagic shock is indispensable. Even if the bleeding is controlled, regeneration of adequate cardio-circulatory functions and microcircular perfusion is essential to secure sufficient organ supply. If control of the bleeding is not possible and blood products are not immediately available, fluid resuscitation under permissive hypotension is inevitable (*Kalkwarf & Cotton, 2017*). We designed our model according to a clinical scenario in which immediate fluid resuscitation is considered lifesaving (*Ziebart et al., 2018*). This study demonstrated that the effect of different infusion regimes and their potential consequences on coagulation are relevant even after several hours. Especially, this long-term effect was an essential key finding of this study: coagulopathy following colloid administration is not only a temporary phenomenon, but traceable for hours after bleeding control and hemodynamic stabilization, but may become crucial with re-occurrence of blood loss or demand for further surgery (*Haas et al., 2008*). In this context, we found a partial recovery four hours after the initial event, which suggests time-dependent progress. On the other hand, experimental factors and the sub-project character limited to address these effects over an even longer time interval to examine full spontaneous recovery. Additionally, we did not retransfuse the removed blood to simulate an early phase without the availability of blood products and assess the mere effects of the different colloids. In this context, specific thrombocyte function analysis or extended coagulation factor determination could deliver additional findings. Regrettably, these were not available for this study. Finally, the comparability of porcine to human physiology allowed for a realistic clinical-like scenario in this study. Notably, however, several physiological differences exist concerning lower hemoglobin content and hypercoagulation in pigs, which must be considered with respect to translational research.

## CONCLUSION

In a porcine hemorrhagic shock model HES, GP, but also standard BES sufficiently stabilized the macrocirculation, but significantly affected coagulation. These effects are most pronounced after colloid and particularly HES administration. The choice of solution in fluid resuscitation affects coagulation, may exacerbate bleeding situation and therefore increase patient's risk. Colloids have to be chosen with caution, because their molecular properties may affect coagulation directly and as a consequence of pronounced hemodilution.

**Institutional abbreviations**

| | |
|---|---|
| **GP** | gelatine-polysuccinate |
| **HES** | hydroxyethyl starch |
| **BES** | balanced electrolyte solution |
| **PiCCO** | pulse contour cardiac output system |
| **SD** | standard deviation |
| **ANOVA** | analysis of variance |

| PTT | activated partial thromboplastin time |
| PT | prothrombin time |
| ROTEM | rotational thromboelastometry |
| CT | clotting time |
| CFT | clot formation time |
| alpha | alpha-angle |
| A10 | amplitude 10 |
| A20 | amplitude 20 |
| MCF | maximum clot firmness |
| MCL | maximum clot lysis |
| HR | heart rate |
| MAP | mean arterial pressure |
| CI | cardiac index |
| Hb | hemoglobin content |
| Hct | hematocrit |
| CI | cardiac index |

## ACKNOWLEDGEMENTS

The study is part of the PhD thesis of Alexander Ziebart. The authors thank Dagmar Dirvonskis for her excellent technical assistance.

### Funding

The study was funded by the German Research Foundation (DFG) ZI 1632/2-1 and Mainz Research School of Translational Biomedicine (TransMed) fellowship affiliated with the Johannes Gutenberg University, Mainz, Germany. The funders had no role in study design, data collection and analysis, decision to publish, or preparation of the manuscript.

### Grant Disclosures

The following grant information was disclosed by the authors:
German Research Foundation (DFG): ZI 1632/2-1.
Mainz Research School of Translational Biomedicine (TransMed).

### Competing Interests

The authors declare there are no competing interests.

### Author Contributions

- Alexander Ziebart conceived and designed the experiments, performed the experiments, analyzed the data, prepared figures and/or tables, authored or reviewed drafts of the paper, and approved the final draft.
- Robert Ruemmler, Christian Möllmann, Jens Kamuf and Andreas Garcia-Bardon performed the experiments, authored or reviewed drafts of the paper, and approved the final draft.

- Serge C Thal conceived and designed the experiments, authored or reviewed drafts of the paper, and approved the final draft.
- Erik K. Hartmann conceived and designed the experiments, analyzed the data, prepared figures and/or tables, authored or reviewed drafts of the paper, and approved the final draft.

## Animal Ethics

The following information was supplied relating to ethical approvals (i.e., approving body and any reference numbers):

This study was conducted after approval by the State and Institutional Animal Care Committee (Chairperson: Dr Silvia Eisch-Wolf; reference number: 23 177-07/G 15-1-092; 01/2016).

## Data Availability

The raw measurements are available in the Supplemental Files.

## Supplemental Information

Supplemental information for this article can be found online at http://dx.doi.org/10.7717/peerj.8399#supplemental-information.

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
