# Peer review of "Fluid resuscitation-related coagulation impairment in a porcine hemorrhagic shock model"

_PeerJ, doi:10.7717/peerj.8399_

## Round 0.1 · original submission · Major Revisions

In the revised manuscript please provide a point-to-point response to reviewers' comments. I would also encourage a short discussion about what the current manuscript adds to the existing knowledge of coagulopathy associated with the administration of colloids in hemorrhagic shock.

·

Basic reporting

The aim of the study and the methods used are straightforward, and importantly the authors' conclusions are justified based on the findings. The manuscript is generally well written and clear; although there are some minor syntax errors; here are a few recommended corrections:

line 69: "Massive blood loss as a hemorrhagic shock trigger requires resolute and efficient action". The word "trigger" needs to be deleted, or re-phrase the sentence.

line 89: "...less frequently applied colloid that.". Delete "that"

line 98: "We thought to explore the influence of..". I would suggest that "We thought" is deleted and this sentence begins with "Our aim was..."

line 99: "...which enables to analyze fibrin polymerization, clot..". Should read "...which enables to fibrin polymerization analysis, clot..."

line 120: delete "their"

line 268: should be changed to read "..can be explained by.." delete the word "with"

Figure 1: define BLH (I assume this refers to baselines)

Figure 1 & 2 axes, Table 1 legend: most of the manuscript provides the units as "kg-1" etc. yet here there's a mix in how the units are expressed e.g. "HR: heart rate (min-1); CI: cardiac index [l/min/m2];" Stick with one way to express things either superscript numbers or /.

Experimental design

The study is well designed, carried out by qualified researchers and in my biased opinion the authors have used an appropriate, human-relevant model to study haemorrhagic shock and resuscitation. All ex-vivo analysis is well described.

I need some clarification on the anaesthesia protocol. It's clearly stated that the pre-medication/sedation is via i.m. ketamine and midazolam; but it's not clear which vein the Fentanyl and propofol where injected prior to intubation. I assume this happened before a central line was inserted. Please clarify.

Validity of the findings

Conclusions are well stated and do not go beyond what the results and methodology provided. The statistics are fine. The comparison with human data was very interesting. As a personal aside, I found the haematocrit data interesting, the low anaemic baseline numbers (when compared to human values) match those we see in our experimental pigs.

Additional comments

I would encourage the authors to keep using large animal models for their research.

·

Basic reporting

See final section

Experimental design

See final section

Validity of the findings

See Final section

Additional comments

I would like to congratulate the authors on a well written and conducted paper.The structure conforms to PeerJ standards. Figures are relevant, of sufficient quality and are well labelled & described. In addition, raw data are supplied.

Although generally very well written, some examples where the language could be improved include lines 220 (not as low), 226 (remained at a higher level), 227 (Regarding clot formation time following GP, only the BES group returned to the baseline level) – the phraseology is sometimes imprecise and so makes the results section a little difficult to follow.

The Introduction & background are well written to show context. However, in recent guidelines, there is an aim to limit early fluid resuscitation and to use crystalloid rather than colloid in the first instance. I have a concern that the premise of this study is based on out-dated practices. While the EMA may have more recently watered down their previous negative risk assessment on use of HES, in practice colloids are not used first line in resuscitation in early haemorrhagic shock. This is due to evidence from human studies. In addition, the coagulopathy noted has been previously well described in humans where experiments using TEG have also been conducted (e.g. Hemostatic changes after crystalloid or colloid fluid administration during major orthopedic surgery: the role of fibrinogen administration. Mittermayr M1, Streif W, Haas T, Fries D, Velik-Salchner C, Klingler A, Oswald E, Bach C, Schnapka-Koepf M, Innerhofer P. Anesth Analg. 2007 Oct;105(4):905-17).

I feel that a much stronger justification for conducting this study needs to be put forward with a more thorough review of the literature that reflects current knowledge and practice. For example, references 13 and 15 do not, in my opinion, support the argument in the text. One is a small study in elective ovarian cancer surgery with goal-directed haemodynamic outcomes. The second showed a worse outcome in ICU patients (increased AKI).

It is stated in the text that “This study represents an independent sub-project and complementary hypothesis of a research project that investigated the cerebral effects of solutions for fluid resuscitation”. It may be reasonable to present these data given that this project is part of a larger project as described. Were it not part of a larger project I would find it difficult to see how ethical approval might be forthcoming given that TEG in haemorrhaging pigs is likely to add little to the already-conducted human studies of TEG in haemorrhaging adults.

The research question is well defined, but, as stated earlier is somewhat less relevant & meaningful than would be expected from primary research in this area. It is stated how the research fills an identified knowledge gap but I am not sure that such a gap has not been explored in human studies. Nonetheless it appears that investigation was performed to a high technical standard. The methods are described with sufficient detail & information to replicate. Conclusions are well stated, linked to original research question & limited to supporting results.

---

## Round 0.2 · accepted · Accept

Dear Dr. Ziebart,

Thank you for your careful attention to the reviewers' critique.
Best wishes with your research!

Kind regards,

Marek Radomski